# Burnout and Resilience among Respiratory Therapy (RT) Students during Clinical Training in Saudi Arabia: A Nationwide Cross-Sectional Study

**DOI:** 10.3390/ijerph192013047

**Published:** 2022-10-11

**Authors:** Rayan A. Siraj, Abdulelah M. Aldhahir, Jaber S. Alqahtani, Hussam M. Almarkhan, Saeed M. Alghamdi, Abdullah A. Alqarni, Munyra Alhotye, Saleh S. Algarni, Fahad H. Alahmadi, Mushabbab A. Alahmari

**Affiliations:** 1Department of Respiratory Therapy, College of Applied Medical Sciences, King Faisal University, Al-Hasa 31982, Saudi Arabia; 2Respiratory Therapy Department, Faculty of Applied Medical Sciences, Jazan University, Jazan 45142, Saudi Arabia; 3Department of Respiratory Care, Prince Sultan Military College of Health Sciences, Dammam 34313, Saudi Arabia; 4Respiratory Care Program, Faculty of Applied Medical Sciences, Umm Al-Qura University, Makkah 21961, Saudi Arabia; 5Department of Respiratory Therapy, Faculty of Medical Rehabilitation Sciences, King Abdulaziz University, Jeddah 22254, Saudi Arabia; 6Department of Respiratory Therapy, King Saud Bin Abdulaziz University for Health Sciences, Riyadh 12271, Saudi Arabia; 7Respiratory Therapy Department, College of Medical Rehabilitation Sciences, Taibah University, Medina 42353, Saudi Arabia; 8Department of Respiratory Therapy, Faculty of Applied Medical Sciences, University of Bisha, Bisha 67714, Saudi Arabia

**Keywords:** burnout, resilience, clinical training, respiratory therapy

## Abstract

Background: The epidemic of burnout has been widely documented among health discipline students. However, there is limited information available on the prevalence of burnout and its association with resilience among clinical-level respiratory therapy (RT) students. Methods: Between March 2022 and May 2022, a descriptive, cross-sectional study using a convenience sample of RT students and interns was conducted. A total of 559 RT students and interns from 15 RT programs responded to socio-demographic questions and the Maslach Burnout Inventory (MBI) and the Brief Resilience Scale (BRS) questionnaires. The data were analyzed using descriptive, inferential, and correlation tests. Results: Of the 559 respondents, 78% reported a high level of burnout. Within the three subscales of burnout, 52% reported emotional exhaustion (EE), 59% reported depersonalization (DP), and 55% reported low personal achievement (PA). The prevalence of burnout increased as students proceeded to senior years (*p* = 0.006). In addition, participants with higher grade point averages (GPA) reported a higher level of burnout. Only 2% of the respondents reported a high level of resiliency. Further, there were negative correlations between resilience and EE (r = −41, *p* < 0.001) and DP (r = −32, *p* = 0.03), and a positive correlation with low PA (r = 0.56, *p* = 0.002). Conclusion: The findings showed a high prevalence of burnout among RT students and interns during clinical training in Saudi Arabia. Resilience was associated with all domains of burnout and is likely to play a protective role. Therefore, there is a need for collaborative interventions to promote resiliency during clinical training to alleviate and overcome burnout symptoms.

## 1. Introduction

Burnout is a psychosocial syndrome initially identified in the 1970s and characterized by three dimensions: emotional exhaustion, depersonalization, and reduced personal accomplishment [1]. Burnout is a significant challenge among healthcare professionals [2], resulting in chronic stress at the workplace and has been associated with mental and physical illnesses [3], medical errors [4], lower empathy, and poor delivery of healthcare services [5].

Respiratory therapists (RTs) are among the healthcare professionals exposed to burnout at the workplace. Recently, 79% of respiratory therapists reported burnout in a survey among 1114 respiratory therapists from 26 medical centers in the United States [6], mainly due to high patient morbidity and mortality as well as heavy workloads in the critical care units [6]. A recent cross-sectional study conducted in a tertiary hospital in Saudi Arabia also reported high levels of the three domains of burnout among RTs: 77% with emotional exhaustion, 98% with depersonalization, and 73% with low personal achievement [7]. 

Students who pursue careers in respiratory therapy may also be prone to burnout. It has been reported that burnout is prevalent among medical [8], dental [9], nursing [10], and physiotherapy students [11]. In addition to the academic stress students may face due to pressure to attain good grades, the fact that RT students and interns observe patient deaths and fear making major mistakes during their clinical rotations may also put them under high pressure—eventually leading to a high level of burnout. There is indeed evidence to show that the severity of burnout increases when students are at clinical levels [9,12], especially when they approach senior years [13]. This suggests that clinical training may be a major factor contributing to burnout among healthcare students. 

Resilience, the ability to bounce back and successfully adjust to stressful events, has recently received more attention because of its positive impacts on health and well-being, as well as its protective role against burnout [12]. Individuals with a high level of resilience may have the ability to overcome the negative effects of the workplace and deal more effectively with stressful situations and increased workloads, which are commonly encountered by healthcare professionals [13,14]. A previous study among US medical students showed that students who reported a high level of resiliency were less likely to experience burnout, stress, and depression [15]. As such, it is important to promote resiliency among RT students and interns. 

To date, no studies have reported on the prevalence of burnout among RT students and interns during their clinical training in Saudi Arabia. Therefore, the aims of this study are to report on the prevalence of burnout and determine the association between burnout and resilience among RT interns and students during clinical training in all respiratory therapy programs across the Kingdom of Saudi Arabia.

## 2. Methods

### 2.1. Study Design and Settings

A cross-sectional questionnaire was used to investigate the prevalence of burnout and resilience and their associations among respiratory therapy students and interns. The data collection process was conducted during the academic year 2021–2022 at 15 respiratory therapy programs (governmental/public and private) across the Kingdom of Saudi Arabia. The respiratory therapy programs were unified and spanned five years. Over the five years, the first year is preclinical, where students mainly study basic science courses, and the remaining years are clinical. The final year (year 5) is the internship, in which students rotate across different hospital settings and work full-time. During the internship year (year 5), interns are only involved in clinical training without any form of schoolwork. Students who were recruited into the study were at the clinical level and free from any health issues (physical and psychological) that would interfere with answering the questionnaire. 

### 2.2. Study Population, Sampling Method, and Data Collection

Using a convenient sampling technique, we approached 800 undergraduate RT students and interns who were at the clinical level. All RT programs across the country were contacted to participate in the study, but only 15 RT programs showed a willingness to participate. Students were excluded if they had not been involved in clinical training in any way, shape, or form. The questionnaires were distributed between 1 March and 31 May 2022. The questionnaire included a cover page where the purpose of the study and the confidentiality of the obtained data were fully explained. Participation in the study was completely voluntary, and students’ consent was obtained. In the case of any further questions, the information of the principal investigator was provided. Participants were asked to fill out the online questionnaire through Monkey Survey (sent via WhatsApp message). Faculty members across the participating RT programs distributed the questionnaires to the students. The faculty sent the anonymous questionnaire to potential participants using a direct link. Once the questionnaire was filled out, respondents were instructed to submit the web form to the web server. The web server then transferred the data into excel for future use. The time taken to fill out the questionnaire was 5 min. 

### 2.3. Instruments

The data were collected for socio-demographic information, the Maslach Burnout Inventory (MBI) [16] and the Brief Resilience Scale (BRS) [17]. The socio-demographic variables included age, gender, marital status, living arrangement, academic level, current grade point average (GPA), length of clinical training (for interns only), and current work shift (for interns). 

### 2.4. Burnout 

Burnout is typically diagnosed based on self-reports. This study used the MBI questionnaire to report on the prevalence of burnout syndrome. The questionnaire comprises 22 questions over three main domains: (1) emotional exhaustion (EE) with nine questions that measure feelings of emotional stress and exhaustion; (2) depersonalization (DP) with five questions that identify individuals with impersonal responses toward their patients; and (3) personal achievement (PA) with eight questions which evaluate lack of success and accomplishment in the clinical training environment. Statements in the MBI questionnaire were modified and adapted to include “clinical training”. To ensure the validity of the questionnaire was maintained, the modified version was piloted with 10 RT students and interns. All of the responses were measured using a seven-point Likert scale intended to assess how frequently RT students have experienced certain feelings identified with their clinical training. The responses on each item ranged from 0 (never) to 6 (every day). The total scores in each of the three items were divided into low, moderate, and high scores. High scores of EE and DP and a low score of PA represented a high level of burnout [18]. 

In this study, burnout was analyzed using two different methods: the sum and the average. The sum method measured burnout at the interval level by adding responses to the MBI items for each subscale and using the sum as the scale score. High burnout was defined as follows: a score of 27 or higher for EE, 10 or higher for DP, and 33 or lower for PA. Moderate burnout was defined as a score of 19–26 for EE, 6–9 for DP, and 34–39 for PA. A score of 18 or lower for EE, 5 or lower for depersonalization, and 40 or higher for PA indicated low burnout [18]. The sum method was used to report on the prevalence of burnout and its subscales. In addition, the average method calculated the mean response for the items that make up each domain separately. The mean scores ranged from 0 (Never) to 6 (Daily), where 0 = never, 1 = a few times a year or less, 2 = once a month or less, 3 = a few times a month, 4 = once a week, 5 = a few times a week, and 6 = every day.

The MBI has been validated in previous research. Internal consistency was estimated using Cronbach’s coefficient alpha and yielded 0.90 for EE, 0.79 for DP, and 0.71 for PA. [16]. These results indicated that the instrument measures the constructs of burnout as intended and that these results across varying and similar populations have proven to be reliable over time. 

### 2.5. Resilience

The Brief Resilience Scale (BRS) was used to measure students’ resilience. The BRS consists of six items, and the responses to each item were on a 5-point Likert scale: 1 = strongly disagree; 2 = disagree; 3 = neutral; 4 = agree; and 5 = strongly agree. Negatively worded items were reverse scored before analyzing and reporting the data. High, normal, and low resilience were defined as scores of 4.3–5.0, 3.0–4.30, and 1.0–2.99, respectively. The BRS instrument has been validated previously [19,20]. The Cronbach’s alpha coefficient of the BRS scale reported elsewhere was 0.83. 

### 2.6. Ethical Consideration

Prior to data collection, ethical approval was obtained from the research ethics committee at King Faisal University, Saudi Arabia (ID: KFU-REC-2022-APR-EA000561).

### 2.7. Statistical Analysis

Data management and analyses were performed using Stata (version 16) statistical software (StataCorp LLC, College Station, TX, USA). Descriptive statistics were calculated and presented as number (%) or arithmetic mean (SD) for categorical and continuous variables, respectively. The normality of the data was assessed to identify appropriate statistical tests. The prevalence of burnout between demographic variables (e.g., male vs. female, high GPAs vs. low GPAs, academic levels, different lengths of clinical training, and work shifts) were compared using the Chi-squared test. The correlation of resilience with burnout subscales (EE, DP, and PA) was determined using Pearson’s correlation coefficient. 

## 3. Results

Of the 800 questionnaires sent to potential participants, 625 (78%) students agreed to participate in the study. After initial screening and accounting for non-response and incomplete questionnaires, 559 (70%) completed questionnaires were identified; and, thus, were included in the analysis, as shown in Figure 1. Male and female students were equally proportional, and more students were attending governmental/public (78%) universities than private (22%) universities. Interns and fourth-year RT students made up more than two-thirds of the total sample. The majority of students lived with their families (82%) and were single (95%). The demographic data are presented in Table 1. 

### 3.1. Prevalence of Burnout

Using the sum method to calculate burnout prevalence, a total of 95% of RT students and interns experienced moderate to high burnout levels. As for the burnout subscales (EE, DP, and PA), participants had mean (SD) scores of 27 for EE (10), 11.7 for DP (7), and 31 for PA (8). In addition, 52% of the RT students and interns were emotionally exhausted, 59% indicated a high level of depersonalization, and 55% indicated a low level of personal accomplishment, as shown in Table 2. 

The prevalence of burnout was also evaluated based on the frequency by calculating the mean responses for the items that made up the burnout subscales. The mean score can take any value from 0 (never) to 6 (every day). The means of the burnout subscales were calculated and revealed the following: EE = 3 (SD 1.1), DP = 2.4 (SD 1.2), and PA = 4 (SD 1). This indicates that, on average, RT students and interns during clinical training experienced emotional exhaustion a few times a month, depersonalization once a month, and a sense of low personal accomplishment at least once a week. 

The prevalence of high burnout experienced by male students (232/279; 83%) was slightly higher than in females (204/280; 73%), although it did not reach statistical significance, *p* = 0.076. Similar results were also observed across the three burnout subscales (EE, DP, and PA). The prevalence of burnout across geographical regions was similar. 

We investigated the variables that were associated with burnout levels and burnout subscales (EE, DP, and PA). As RT students advanced their academic level, they experienced high levels of burnout, as shown in Figure 2. Further, 4th year RT students and interns showed increased levels of burnout (*p* = 0.006), emotional exhaustion (*p* = 0.038), and depersonalization (*p* = 0.048) compared to students in 1st, 2nd, and 3rd years.

The prevalence of burnout and its subscales were also evaluated based on students’ GPAs. Students with high GPAs (4.51–5.0) had a high burnout level (175/319; 55%) compared to students with lower GPAs, *p* = 0.003. Similarly, those students (with higher GPAs) were emotionally exhausted (*p* = 0.017) and showed a sense of low accomplishment (*p* = 0.047) compared with their peers with low GPAs. 

Within interns (n = 190), those who were longer than six months into clinical training (n = 156/190) showed signs of high burnout syndrome (85%), emotional exhaustion (56%), depersonalization (67%) and low personal accomplishment (55%) compared to interns who were less than six months into clinical training, despite not reaching a statistical significance (*p* > 0.05). We found no difference between current shift work (day or night) in terms of burnout levels, EE, DP, or PA. 

### 3.2. Resilience

The mean (SD) score of resilience was 3.1 (0.5), indicating a normal resilience level. As shown in Table 3, a third of the sample population experienced low resilience, while only a small proportion showed high resilience, 2%. The majority of RT students showed a normal resilience level. There were no differences found in terms of resilience levels across demographic variables. 

### 3.3. Associations between Resilience and Burnout Subscales (EE, DP, and PA) 

There were negative correlations between resilience score and emotional exhaustion (r = −0.41, *p* < 0.001) and depersonalization (r = −0.32, *p* = 0.03). A positive correlation was found between resilience and personal accomplishment (r = 0.46, *p* < 0.001). 

## 4. Discussion

In Saudi Arabia, this is the first study to report on burnout syndrome and its relationship with resilience among RT students and interns. The findings of the current study revealed that RT students and interns experienced moderate to severe burnout, and burnout was more prevalent as students advanced in their academic levels. Moreover, resilience was significantly associated with the three domains of burnout (EE, DP, and PA); and only a small percentage of RT students and interns showed a high level of resiliency. 

Burnout syndrome presents a significant issue for healthcare professionals [2] as well as for students who wish to pursue an RT career in healthcare. In general, burnout is a common syndrome across health discipline students. A meta-analysis of 12 studies reported that up to 75.2% of medical students are burnt out [8]. In line with this, we found that 78% of RT students and interns experienced severe burnout during clinical rotations. This may be explained by the fact that RT students and interns are exposed to numerous stressors, such as competitive entrance GPAs, dealing with a tremendous amount of information over short periods, and/or preparation for the Saudi Commission for Health Specialties exam (this exam determines whether students/interns are allowed to practice after graduation). More importantly, there is the fear of making mistakes in a clinical setting, especially when interacting with critically ill patients under students’ care. All of these factors may contribute to increased levels of stress and, ultimately, burnout. 

In this study, we found that the prevalence of burnout increased as students advanced in their studies. This is concordant with previous studies conducted among medical students [21,22]. It seems logical that this effect may be due to the significant impact of clinical exposures encountered by the students, especially when they proceed to senior years [23]. Indeed, RT students have to attend several hours of classes per week in addition to their clinical training hours, which happens to be mentally and physically exhausting. During the internship period, moreover, students are expected to have clinical duties similar to a full-time employee, such as having a heavy clinical load, managing 12-h shifts (day and night), and rotating across clinical areas. This transition (from working under supervision to working with less or even no supervision) is likely to put students under significant stress and contribute to increased levels of burnout. 

Our findings that students and interns with high academic performance (measured by GPA) experienced higher degrees of burnout compared to those with lower GPAs are worth discussing. It is likely that those individuals (with higher GPAs) put more time, effort, and energy into their coursework to achieve good grades and are also pressured by their families to always be “an A student”. It is also possible that those students seek competitive opportunities after graduation, such as acquiring a government scholarship to continue their education, getting a job in academia, or even working as a full-time therapist in a very reputable hospital in the country. Seeking such opportunities (which require competitive GPAs) puts students under extreme stress, which then translates to high levels of burnout. 

When each dimension of the burnout scale was assessed separately, we found that more than half of the participants reported a high level of emotional exhaustion (52%) and depersonalization (59%) and a low level of personal achievement (55%); findings that are similar to studies conducted among students of other health disciplines [20,24,25]. Da Silvia et al. demonstrated a high level of emotional exhaustion and depersonalization and a low level of professional efficacy among nursing students across three universities in Brazil [25]. This indicates that a higher level of burnout is associated with a lower level of personal achievement, as has been reported previously [26]. Thus, professional effectiveness among RT students and interns is likely to compensate for academic life stress. 

Although burnout is a tridimensional syndrome, it is nevertheless important to take into count each dimension separately. Emotional exhaustion, which was shown to be highly prevalent in our study, is thought to be the initial feature of burnout and an indicator of mental depletion. A previous 3-year longitudinal study of 1702 nursing students reported that the level of emotional exhaustion increased with time and was associated with an increased prevalence of depression and life dissatisfaction [27]. Moreover, students with a low level of personal achievement (professional efficacy), as shown in our findings, were likely to feel that they were incompetent and unqualified and, consequently, may seriously consider leaving the RT program. There was also evidence to suggest that low scores for personal achievement were linked with depressive symptoms, serious thoughts of dropping out, and a lower level of quality of life [28]. Therefore, strategies are needed to minimize the prevalence of burnout in general and emotional exhaustion and low personal accomplishment in particular. 

The role of resilience in terms of facilitating a healthy and more productive workplace among healthcare professionals has been highlighted [29,30]. In addition, being highly resilient allows for the effective management of stressful situations and heavy workloads [13,14], such as that encountered by RT students and interns during their clinical rotations. In this study, we found that only a small proportion of students’ scores reached the standard of high resilience. Although this may seem a little surprising, it is nevertheless expected, given the high prevalence of burnout reported by students. Furthermore, resilience was significantly associated with the three domains of burnout (EE, DP, and PA). This indicates that training programs or preceptorship programs must consider monitoring emotional status related to the RT profession. This can be completed using a structured preceptorship program that considers the cultural and gender differences during training sessions [31]. Another explanation for these results can be that the progression in the RT training and/or well-prepared preceptorship program is responsible for increasing self-confidence as well as promoting resiliency [5]. 

### Strength and Limitations

A strength of this study is that it is the first and only study conducted in Saudi Arabia to explore such issues among RT students and interns. Secondly, participants were recruited from 15 different universities and colleges across the country, thus, offering high external validity. However, this study has some limitations. Results from this study should be described with caution as students have faced the COVID-19 pandemic and the RT profession was one of the front-line professions with a high workload. Secondly, the cross-sectional nature of the study did not allow for the assessment of any causality. In addition, we were unable to assess the perceptions of those with access to preceptorship programs during RT training or pre-employment programs, as these programs can help minimize burnout and promote resiliency. 

## 5. Conclusions

The burnout rate among RT students and interns is moderate to severe, and students are more likely to experience burnout as they progress through their academic levels. Only a small percentage of students showed high resiliency, and there was a significant correlation between resilience and the three domains of burnout (EE, DP, and PA). Effective interventions for improving resilience and reliving burnout should be implemented, along with facilitating a healthy environment for clinical training. 

## Figures and Tables

**Figure 1 ijerph-19-13047-f001:**
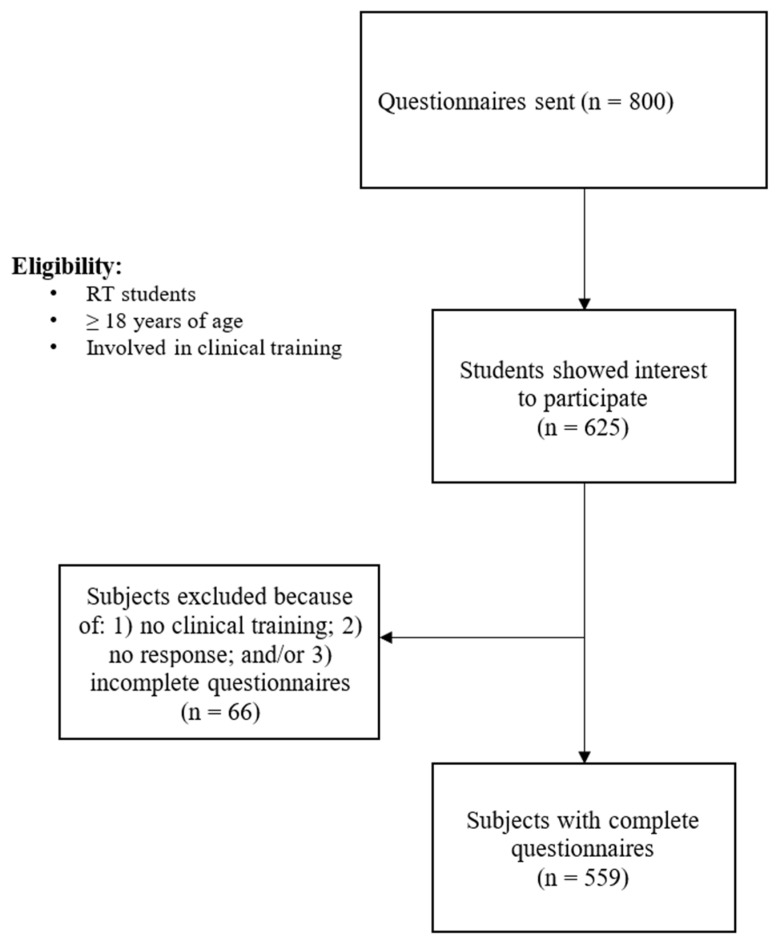
Flow chart of the study.

**Figure 2 ijerph-19-13047-f002:**
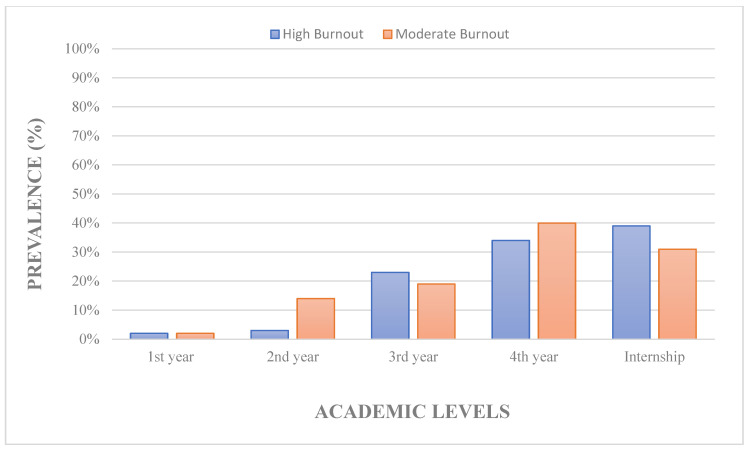
Prevalence of RT students and interns experiencing burnout across academic levels.

**Table 1 ijerph-19-13047-t001:** Demographic data of study participants (n = 559).

Variable	
Age, years (Mean (SD))	22 (2)
Gender (male %)	279 (50%)
University sector, n (%)	
Governmental (public)	436 (78%)
Private	123 (22%)
Geographical Region, n (%)	
Eastern	140 (25%)
Central	124 (22%)
Western	110 (19%)
Southern	103 (18%)
Northern	82 (14%)
Academic level, n (%)	
1st year	17 (3%)
2nd year	28 (5%)
3rd year	134 (24%)
4th year	190 (34%)
Intern	190 (34%)
Cumulative GPA, n (%)	
4.51–5.00	319 (57%)
3.51–4.50	190 (34%)
3.0–3.51	34 (6%)
<3.00	16 (3%)
Living arrangements, n (%)	
Living with family	458 (82%)
Living elsewhere	101 (18%)
Marital status, n (%)	
Single	531 (95%)
Married	27 (4.80%)
Divorced or widowed	1 (0.20%)
Length into the internship (interns only), n (%)	
0–3 months	13 (7%)
4–6 months	21 (11%)
7–9 months	59 (31%)
More than 9 months	97 (51%)
Current shift work (for interns only), n (%)	
Day	76 (40%)
Night	114 (60%)

Data are presented as n (%) or mean (standard deviation (SD)). Abbreviation: GPA, Grade Point Average.

**Table 2 ijerph-19-13047-t002:** Prevalence of burnout among RT students and interns during clinical training (n = 559).

Burnout	N (%)
High	436 (78%)
Moderate	95 (17%)
low	28 (5%)
**Burnout subscale**	
Emotional exhaustion	
High	291 (52%)
Moderate	129 (23%)
Low	139 (25%)
Depersonalization	
High	330 (59%)
Moderate	117 (21%)
Low	112 (20%)
Personal accomplishment	
Low	307 (55%)
Moderate	139 (25%)
High	112 (20%)

Results are presented as frequency (%) unless stated otherwise.

**Table 3 ijerph-19-13047-t003:** Resilience level.

Resilience, Mean (SD)	3.1 (0.1)
High (4.3–5.0)	11 (2%)
Normal (3.0–4.30)	369 (66%)
Low (1.0–2.99)	179 (32%)

Results are presented as frequency (%) unless stated otherwise.

## Data Availability

The data presented in this study are available on reasonable request from the corresponding author.

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
