# Peer review of "Burnout and Resilience among Respiratory Therapy (RT) Students during Clinical Training in Saudi Arabia: A Nationwide Cross-Sectional Study"

_ijerph, 2022, doi:10.3390/ijerph192013047_

Round 1

Reviewer 1 Report

The authors reported new results on burnout and resilience among students and interns during their clinical training in respiratory therapy facilities in Saudi Arabia.

Their results corroborate with similar findings in other fields of clinical training.

The reported results emphasize problems related to burnout and resilience among medical students/interns and would be of interest to understand the problem and to develop methods to improve the process of clinical training.

I have a few minor comments, which, I believe, could improve the quality of presentation.

(1) Abstract, Lines 27 - 30. The authors indicated that "A total of 800 RT students and interns from 15 RT programs were invited ..." (lines 27-28), but only 559 individuals were included into the study.

These two numbers could confuse the reader. I would suggest to rewrite this and related parts by retaining only the number of actual participants, which is 559, in this study. The number of students/interns invited to participate is not relevant to this study.

(2) Table 1. Could the authors introduce a short description of "SD"?

(3) Lines 202-203. The authors provided p-values, p=0.076, when comparing prevalence of high burnout in males vs females.

The authors did not mention any method that was used for this estimation.

Could the authors provide a description of the respective method(s) in the Methods section?

(4) Figure 2. The type of figure (continuous variables) is not appropriate for categorical variables.

I would suggest to present results with more appropriate plot types, for instance, bars or scatter plot with confidence intervals.

(5) Lines 219-223. The authors presented a general description, but not results. I would suggest either removing this paragraph or presenting respective evidence/results.

These results could be displayed either in the main text or in supplementary materials.

Author Response

We greatly appreciate the reviews by the Chief Editor and Reviewers. We have revised our paper in the light of the useful comments, and we hope we have addressed all concerns indicated in the review report.

Response to Comments from Reviewer 1

Comment I:

Abstract, Lines 27 - 30. The authors indicated that "A total of 800 RT students and interns from 15 RT programs were invited ..." (lines 27-28), but only 559 individuals were included into the study.

These two numbers could confuse the reader. I would suggest to rewrite this and related parts by retaining only the number of actual participants, which is 559, in this study. The number of students/interns invited to participate is not relevant to this study.

Response I:

Thank you for this comment. We have addressed this point, and it now reads as follows “ A total of 559 RT students and interns from 15 RT programs responded to the sociodemographic, Maslach Burnout Inventory (MBI), and Brief Resilience Scale (BRS) questionnaires”

Comment II:

Table 1. Could the authors introduce a short description of "SD"?

 Response II:

This has been addressed in Table 1.

Comment III:

Lines 202-203. The authors provided p-values, p=0.076, when comparing prevalence of high burnout in males vs females.

The authors did not mention any method that was used for this estimation.

Could the authors provide a description of the respective method(s) in the Methods section?

Response III:

Thank you for pointing that out. We revised the previous sentence to provide more clarity. It now reads as follows (lines 163-166) “The prevalence of burnout between demographic variables (e.g. male vs female, high GPAs vs low GPAs, academic levels, different length of clinical training, and work shift) were compared using Chi-squared test”.  

Comment IV:

Figure 2. The type of figure (continuous variables) is not appropriate for categorical variables.

I would suggest to present results with more appropriate plot types, for instance, bars or scatter plot with confidence intervals.

Response IV:

Thank you. We have addressed this in accordance to the reviewer’s suggestion.

Comment V:

Lines 219-223. The authors presented a general description, but not results. I would suggest either removing this paragraph or presenting respective evidence/results.

These results could be displayed either in the main text or in supplementary materials

Response V:

Thank you. We provided numbers (%) to this paragraph (lines 2019-223). It now reads as follows” Within intern students (n=190), those who have been longer than six months into clinical training (n=156/190) showed signs of high burnout syndrome (85%), emotional exhaustion (56%), depersonalization (67%) and low personal accomplishment (55%) compared to interns who have been less than six months into clinical training, despite not reaching statistical significance (p>0.05)”.  

Reviewer 2 Report

Dear Authors.

This paper is very well written and proven statistically.

Yet, I have several suggestions to take into consideration.

Table 1.

“Governmental” equals “state” or not?

Line 201 and further: my concern is whether the interns and last year students should be classified together. [Is it possible to be an intern for four years? (line 208) – in Your questionnaire there is a position >9 months]

What about the nationality, cultural differences of participants? You have mentioned future directions in lines 306-307 basis on “cultural differences”, yet, You did not indicate any in Your study.

Burnout ratios – it would be interesting to compare them to other sectors (f.e. industry, financial ones). Maybe the level of stress, determining burnout, is typical? And the burnout is not a fluctuation but the normal state related to the work environment? I strongly recommend to analyze the Rule of Yerkes-Dodson.

Kind Regards

Reviewer

Author Response

(The authors gave the same response as above.)
